# The Effect of Long-Term Agricultural Use on the Bacterial Microbiota of Chernozems of the Forest-Steppe Zone

Konstantin S. Boyarshin [1], Valeria V. Adamova [2,*], Zheng Wentao [1], Olga Y. Obuhova [1], Marina V. Kolkova [1], Vitaliy A. Nesterenko [3], Olga S. Bespalova [1], Violetta V. Kluyeva [1], Kristina A. Degtyareva [1], Yulia N. Kurkina [1], Olesia A. Makanina [1] and Irina V. Batlutskaya [1]

[1] International Laboratory for Applied Biotechnology, Institute of Pharmacy, Chemistry and Biology, Belgorod State University, Belgorod 308015, Russia

[2] Department of Biology, Institute of Pharmacy, Chemistry and Biology, Belgorod State University, Belgorod 308015, Russia

[3] V.V. Dokuchaev Soil Science Institute, Moscow 119017, Russia

[*] Correspondence: adamova_v@bsu.edu.ru; Tel.: +7-(903)-884-80-74

**Abstract:** The structure of soil microbial communities and the factors that control it are still poorly understood and cause ongoing interest. The diversity of soil bacteria reflects the diversity of existing ecological niches and trophic connections between them and with other components of the ecosystem. The presence of certain taxa with their own characteristic properties depends on the specific environmental conditions. Analysis of the composition of soil microbiota in various physicochemical conditions allows identify landmarks for understanding the principles by which it is formed. Of particular interest in this regard are the features of cultivated fertile soils that assist agricultural production. In this paper, we have assessed the occurrence of representatives of different families of bacteria in arable and nonarable chernozems of three subtypes. The methodology of 16S microbial profiling was used. The general features of the taxonomic structure of bacterial communities of chernozem remain similar, with a high occurrence of the families *Sphingomonadaceae, Xanthobacteraceae, Rubrobacteraceae* and *Chitinophagaceae*. Notably, *Gemmatimonadaceae*, one of the most commonly occurring families, is approximately twice as represented in arable soils as in nonarable ones. Differences between subtypes of chernozem and between arable and nonarable areas concerned different sets of bacterial families. Among others, the occurrence of representatives of families characterized by nitrogen fixation, nitrite oxidation and reduction, ethanol oxidation, biodegradation and microbial predation is touched upon. The obtained results raise the question of the factors limiting the number of certain groups of bacteria in various soil conditions and so limiting their contribution to the properties of the ecosystem.

**Keywords:** soil microbiota; chernozem; 16S microbial profiling

## 1. Introduction

Chernozems, also known as Mollisols, Black Soils or Prairie Soils, are formed mostly on loess sediments in moderately moistened continental areas under grassland biomes in the middle latitudes, occupying about 916 million ha, or 7% of the ice-free surface of the Earth [1]. They are characterized by high content of humus, calcium, potassium, phosphates [2] and pH values near neutral [3]. High concentrations of ion-binding [4] and moisture-preserving humic substances ensure soil fertility [5], contributing to the conservation and usability of introduced nutrients.

The chemical composition of chernozems, rich in organic and mineral substances, contributes to the maintenance of a variety of ecological niches occupied by bacterial microbiota. The bacterial branch of life has a special role in ensuring soil fertility, including participation in the synthesis of humic substances, fixation and oxidation of nitrogen, dissolution of phosphates, suppression of pathogens and symbiotic interactions with the root system of

plants [6]. Soil bacterial communities are connected by syntrophic, competitive and other synergistic and antagonistic interactions within themselves and with other components of the biota. Their taxonomic composition reflects the ecological balance resulting from the combined effects of biotic [7] and abiotic [8,9] factors, including controllable factors related to agricultural technology. The idea of controllability of processes occurring in the soil through the introduction of biofertilizers, stimulation or suppression of individual bacterial groups has far-reaching prospects.

Mechanized arable farming has a special effect on soils. Regular harvesting and fertilization determine the balance of nutrients and organic matter. The decomposition of organic fertilizers causes an acidification effect [10], which is repaid or not repaid by liming measures. Mechanical ramming of the soil by the wheels of machinery changes the structure of the soil, making it difficult to aerate and drain it. Soil biota is influenced by the use of pesticides and other agrochemicals.

Data on different types of soils, characterizing the number and diversity of bacteria during agricultural processing and without it, show different magnitude and direction of changes. Study on Fluvo-Aquic soils had shown only small differences in abundances of 17 considered bacterial phyla in soils objected to common and rotary tillage and without tillage. OTU richness and equitability values also remained similar [11]. Sandy loam soil at a depth of 10–20 cm was shown to have a significantly higher concentration of heterotrophic bacteria in the conditions of ordinary plowing in comparison with moderate plowing and its absence. Shannon diversity coefficient was somewhat higher for any tilled ground if compared with untilled ones [12]. However, on Decatur silt loam soil under the long-term no-till treatment, rising bacterial diversity was shown by phospholipid fatty acid analysis [13]. At the same time, despite the small scale of the changes, it was shown that even reduced soil disturbance by tillage in agricultural fields may have impact on soil biota as close as intensive tillage does [14]. Even a single tillage event may cause remarkable changes in it [15].

In Voronic (or typical) Chernozem, using differential cultivation, it was shown that the number of ammonifying bacteria in conditions of regular plowing is significantly higher than in conditions of virgin land and especially in conditions of fallow land [16]. Analysis of taxonomic composition by molecular profiling shows an increased diversity of bacteria in the subsurface layer of arable chernozem in comparison with fallow soil and forest [17]. *Actinobacteria*, *Proteobacteria*, *Acidobacteria*, *Bacteroidetes*, *Verrucomicrobia*, *Gemmatimonadetes*, *Chloroflexi*, *Firmicutes* and *Planctomycetes* phyla dominate the bacterial communities in Voronic Chernozem [17,18], as well as *Nitrospirae* [17] and *Cyanobacteria* [18]. On arable plots drastic decrease of *Verrucomicrobia*, represented mainly by genus *Chthoniobacter*, and significant decrease of genus *Rhodoplanes* in *Pseudomonadota* phylum were shown [17].

In another study on Voronic Chernozem, *Nitrospiraceae* phylum reduced its share by more than twofold in arable soil [18]. All the differences presented in the work in the microflora of arable and nonarable samples demonstrate a more than twofold drop in the share of every of differing taxa in arable ones. This is how 11 families and 13 genera behave; however, none of them exceeds 0.4% of the bacterial microbiota [18].

These presented works on the microflora of chernozem [17,18] are generally consistent in listing the dominating phyla, but do not converge in describing the changes associated with arable farming. Unfortunately, the small number of such works makes it difficult to conduct meaningful meta-analysis. At the same time, such differences, concretized at the level of families, genera and species, are of deep interest for understanding the processes occurring in arable soils.

We rely on the hypothesis that the taxonomic composition of the bacterial soil microbiota at its lower levels largely reflects the cumulative effect of ecological factors including agricultural use and directs the concomitant changes in soil properties [19–21]. Thus, the study of the general patterns of microbiota composition in arable and nonarable conditions for a type of soil lays the foundation for a detailed study of the mechanisms of processes driven by intensive farming. Among them are soil erosion effects, such as acidification and

humus loss [22,23], obligatorily interconnected with functioning of microbiota. Moreover, identification of characteristic features of bacterial communities of arable soils can serve understanding of the ecological roles of certain common bacterial species, many of which are still poorly studied. In turn, deepening the understanding of bacterial life in soil can show new possibilities for soil fertility management.

The forest-steppe zone of eastern Europe is an important agricultural area characterized by both high soil fertility and a sufficient level of its natural moisture. In this study, we used three subtypes of chernozem, the most characteristic of this zone, namely Voronic, Vorony-Calcic and Grey-Luvic Chernozems, to determine the average trend of microbiota changes during long-term agricultural use. On this material, using the 16S microbial profiling method, we expected to obtain statistically confirmed differences at the level of families and, if possible, at the level of genera, allowing us to draw some microbiological conclusions for chernozem soils of this climatic zone and discuss the characteristic ecological features of the groups subject to changes.

## 2. Materials and Methods

### 2.1. Sampling Sites and Sample Collection

This study was conducted in the forest-steppe zone of eastern Europe, in the Belgorod region of Russia (Figure 1). In this area, the average annual temperature ranges from +5.4 to +6.8 °C, the duration of the frost-free period average is determined from 155 to 160 days, the summer soil temperature at depths of 40 cm is close to 20 °C, the average annual precipitation is 627 mm, of which 68 mm falls in the wettest month of June. Soil samples were collected in the range of latitudes 50.21074–51.18853 and longitudes 36.96127–38.78922.

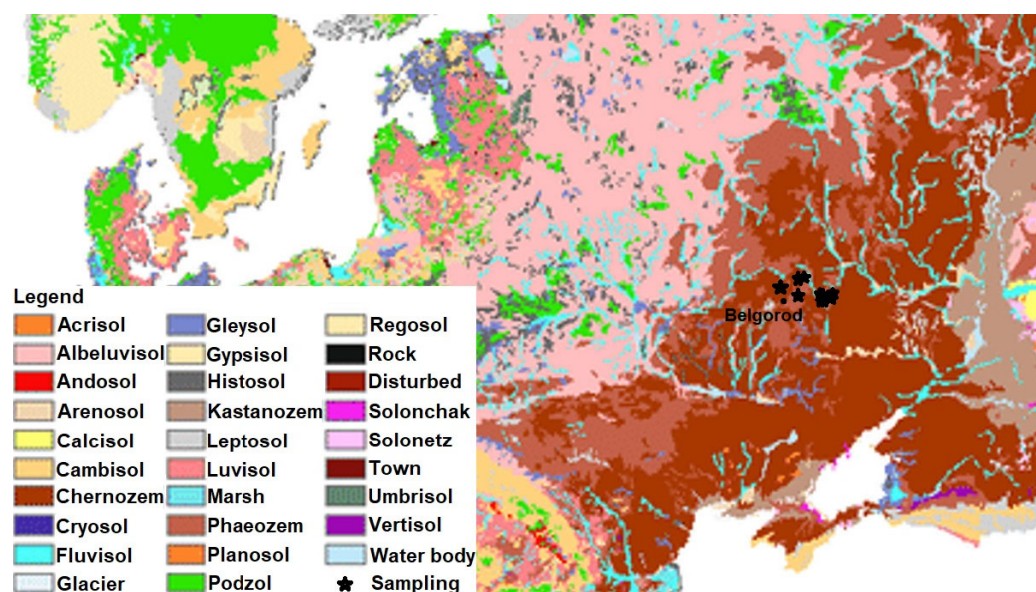

**Figure 1.** Sampling area on the soil map of eastern Europe.

The sampling sites were selected on the elevations or top of their gentle slopes between river valleys and ravines. Sampling was carried out in the second half of June, before the end of the active vegetation of winter wheat on arable sites and of the grass cover on non-arable ones. Linear sections of fields 30–50 m in length sown with winter wheat and adjacent non-arable areas covered with grasses or forest plantations were used for sampling. Samples were taken from a depth of 10–20 cm. One combined sample was taken from 10–12 points spaced 3–6 m apart and characterized one plot. The obtained 1–1.5 kg of soil was placed in a tightly tied plastic bag and frozen at −80 °C on the day of sampling.

In such manner, samples of each soil subtype, Vorony-Calcic Chernozems, Voronic Chernozems and Grey-Luvic Chernozems were taken to compare the composition of

bacterial communities (Table 1). The subtypes of chernozem were determined by Rusagro Invest company, which cultivates these lands. Every arable sampling area had its paired nonarable site at a distance of 10 to 20 m. At the same time, the distance between such pairs for one subtype of chernozem ranged from 22 to 92 km. Additional pair of samples of Voronic Chernozem was taken on both sides of the border of Yamskaya Steppe area of Belogorye State Nature Reserve.

**Table 1.** Number of combined soil samples from arable and non-arable sites belonging to Voronic, Vorony-Calcic and Grey-Luvic subtypes of Chernozem.

|  | Voronic | Vor.-Calcic | Grey-Luvic | Total |
|---|---|---|---|---|
| Arable | 4 | 3 | 3 | 10 |
| Nonarable | 4 | 3 | 3 | 10 |
| Total | 8 | 6 | 6 | 20 |

### 2.2. Microbiological Profiling

Total soil DNA was isolated using FastDNA Spin Kit for Soil DNA Extraction (MP Biomedicals, Santa Ana, CA, USA). The samples were prepared using two-stage PCR. At the first stage, the amplification of the hypervariable V3–V4 region of the 16S rRNA gene was performed using primers 5′-CCTACGGGNGGCWGCAG-3′ and 5′-GACTACHVGGGTATCTAATCC-3′ with degenerate nucleotide sequences universal for all bacteria. At the second stage, PCR amplification of the product obtained at the first stage was performed in order to barcode the library. The amplicons obtained after purification on magnetic particles and concentration measurement by the fluorimetric method were ready-made DNA libraries for multiplex sequencing. DNA analysis was carried out on Illumina MiSeq sequencer by the paired-end reading method generating at least 10,000 paired readings per sample.

Sequencing data processing was carried out using the automated QIIME 1.9.1 algorithm [24], which includes combining forward and reverse readings, removing technical sequences, filtering sequences with low reading reliability of individual nucleotides (quality less than Q30), filtering chimeric sequences, alignment of readings to the reference 16S rRNA sequence, sequence distribution by taxonomic units using the Silva database [25] version 132. The algorithm of classification of open-reference operational taxonomic units (OTU) with classification threshold of 97% was used.

### 2.3. Statistical Analysis of the Results

Data analysis was performed in R version 4.1.2 [26] using the vegan package [27]. Permutational multivariate analysis [28] of variation using Bray–Curtis distance matrices [29] was used to assess the influence of land use and soil type on the community structure at the family level. Indicators of alpha beta and gamma biodiversity [30], Pielou index of evenness [31] and family richness were calculated for every sample. We applied beta diversity metrics that use presence–absence data [30,32] and calculated beta diversity as a function of alpha and gamma diversity [33] with the Sørensen index of dissimilarity [34].

For every family, a Kruskal–Wallis test [35] and pairwise comparisons using Wilcoxon rank sum exact test [36] with Bonferroni correction were performed to assess the significance of the differences of its abundance in different types of Chernozem and for arable and nonarable conditions.

## 3. Results
### 3.1. Comparison of Biodiversity Indicators

Classification of the majority of OTUs was carried out at the level of families; less than half were classified up to the genera level. Thus, the main statistical analysis of the composition of bacterial communities was carried out at the family level. According to the results obtained at this level, family richness tends to be at a comparable level for all studied samples (Figure 2).

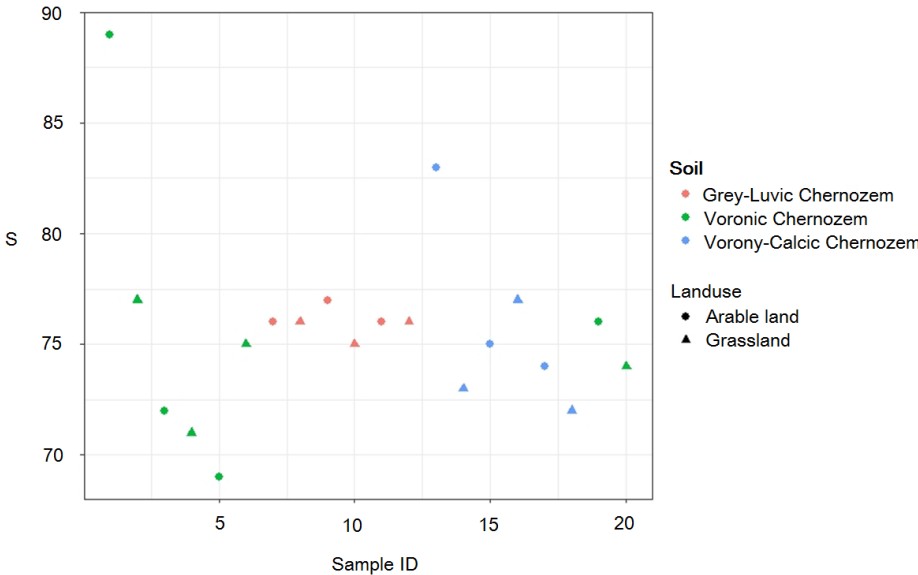

**Figure 2.** Family richness of the samples' scatter plot. Richness values are deferred on the vertical axis. Nonarable sites marked as grassland. Odd sample IDs and even ones to the right of them correspond to paired samples.

The largest values characterize the non-arable Gray-Luvic and Vorony-Calcic Chernozems, the smallest arable Gray-Luvic Chernozems. The Pielou evenness index for most samples also remains at a similar level (Figure 3), although for some of them its values may deviate significantly, mainly highlighting more even communities.

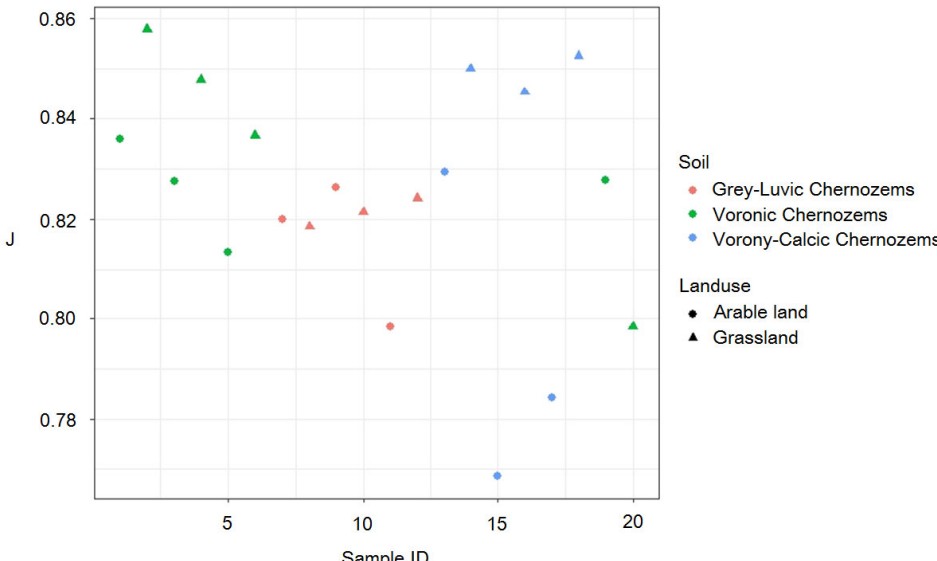

**Figure 3.** Pielou evenness indexes scatter plot. Index values are deferred on the vertical axis. Nonarable sites marked as grassland. Odd sample IDs and even ones to the right of them correspond to paired samples.

Alpha diversity tends to be slightly greater in arable soils and beta diversity in non-arable ones (Table 2). For three examined soil subtypes, Grey-Luvic Chernozem is characterized by the highest alpha diversity and Voronic Chernozem by the highest beta diversity.

**Table 2.** Indexes of diversity for arable and nonarable sites and for three subtypes of chernozem.

| Factors | | Diversity | | |
|---|---|---|---|---|
| | | **Alfa** | **Beta** | **Gamma** |
| Land use | Arable land | 76.7 | 0.186 | 91 |
| | Grassland | 74.6 | 0.206 | 90 |
| Soil subtype | Grey-Luvic chernozems | 76 | 0.184 | 90 |
| | Vorony-Calcic Chernozems | 75.7 | 0.189 | 90 |
| | Voronic Chernozems | 75.4 | 0.207 | 91 |

### 3.2. Comparison of Taxonomic Structure of the Communities

All the studied communities are characterized by a pronounced dominance of proteobacteria and actinobacteria and demonstrate similar shares of the main phyla (Figure 4). Occurrence of 91 families (Figure S1) and 159 genera (Figure S2) was analyzed. On the level of families, *Sphingomonadaceae* (1.61–9.77%), *Xanthobacteraceae* (2.54–8.85%), *Gemmatimonadaceae* (1.68–10.19%), *Chitinophagaceae* (3.17–6.65%) and *Rubrobacteriaceae* (1.41–4.96%) tend to dominate others. In the *Sphingomonadaceae*, genus *Sphingomonas* (1.03–8.84%) and in *Xanthobacteraceae, Bradyrhizobium* (0.78–3.51%) make up the majority or a significant part. Among *Gemmatimonadaceae, Gemmatimonas* (0.31–5.80%), as well as *Rubrobacter* (1.19–4.96%) among *Rubrobacteriaceae* are the most numerous. In *Chitinophagaceae* genus, *Flavisolibacter* (0.20–3.44%) should be noted, although in this family other genera are also able to have occurrence above one percent.

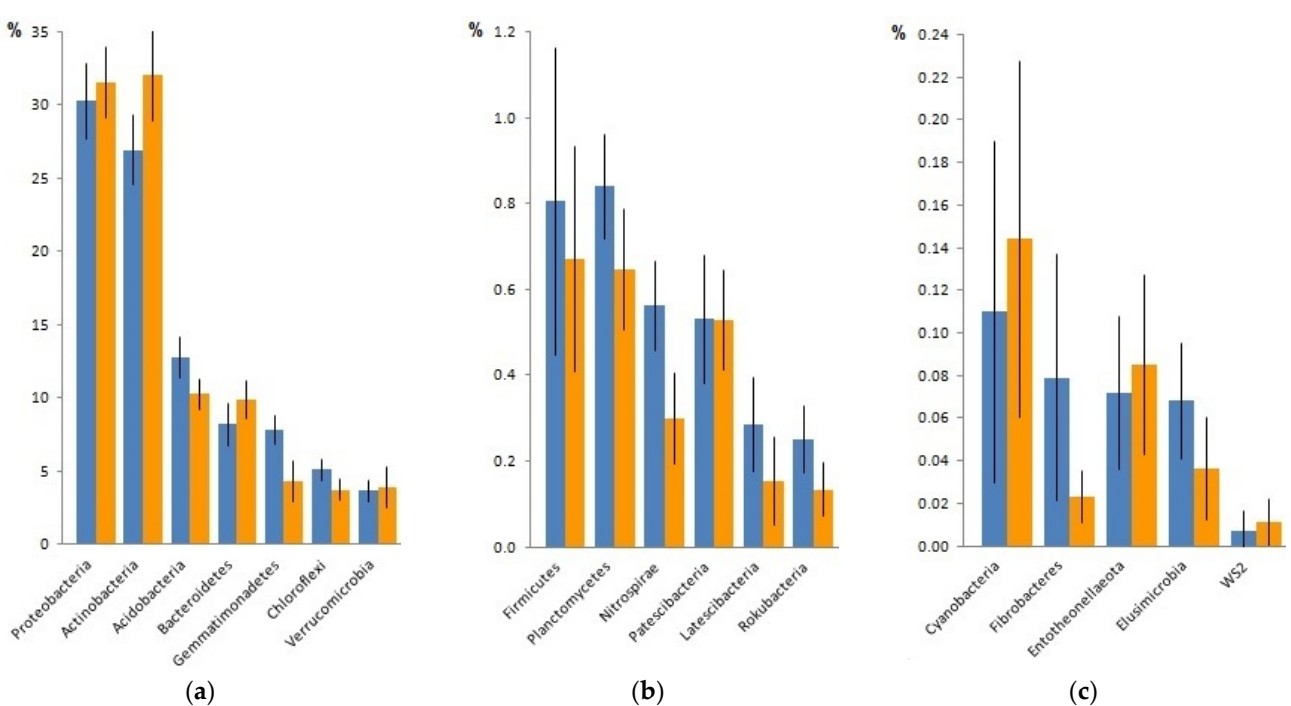

(a)  (b)  (c)

**Figure 4.** Taxonomical structure of chernozem bacterial communities on the level of phyla, (**a**–**c**) include different phyla and differ in scale. Blue bars represent arable soils, orange represent non-arable. Confidence intervals at 95% showed as error bars were calculated for *n* = 10.

According to the results of permutational multivariate analysis, factors of land use and soil subtype act on the structure of the bacterial community predominantly separately (Table 3).

**Table 3.** Permutational multivariate analysis of influence of agricultural usage (Land use) and Chernozem subtype (Soil) on the structure of soil communities.

| Factors | Df | Sum Of Sqs | R² | F | Pr(>F) |
|---|---|---|---|---|---|
| Land use | 1 | 0.14431 | 0.16008 | 4.0736 | 0.002 |
| Soil | 2 | 0.20822 | 0.23097 | 2.9388 | 0.001 |
| Land use:soil | 2 | 0.05302 | 0.05881 | 0.7482 | 0.720 |
| Residual | 14 | 0.49597 | 0.55015 | | |
| Total | 19 | 0.90152 | 1.00000 | | |

Factor of the land use establishes near 16% of community variation. Comparing the average values characterizing ten samples of arable soils and ten samples of nonarable soils using Wilcoxon rank sum criterion with Bonferroni correction, without taking into account the soil subtype, 20 families that differ were identified (Table 3). They belong to seven phyla with majority belonging to dominating *Proteobacteria* and *Actinobacteria*. The most numerous of them are *Gemmatimonadaceae*, making greater fraction in arable grounds.

By the same method families whose shares differ in chernozems of different subtypes have been identified (Table 4). Despite rather big imply of the soil subtype (Table 2) these differences concern only two main phyla, *Proteobacteria* and *Actinobacteria*. Noteworthy is the significant difference between Voronic and the other two investigated Chernozem subtypes, without near any difference between these two. The most numerous of the differing families are *Sphingomonadaceae* showing greater percentages in Vorony-Calcic and Grey-Luvic chernozems (Table 5). In general, the differences found in the taxonomic structure of communities depending on the type of soil or agricultural use require discussion with the involvement of data on the specific features of individual affected families and genera.

**Table 4.** Average percentage of families that differs in their share in arable (A.) and nonarable (N.a.) grounds. Bigger values given in bold.

| Phyla (Class) | Familia | %A. | %N.a. | *p* |
|---|---|---|---|---|
| | *Reyranellaceae* | 0.44 | **0.71** | 0.016 |
| (*Alphaproteobacteria*) | *Beijerinckiaceae* | 0.76 | **1.13** | 0.041 |
| | *Paracaedibacteraceae* | **0.02** | 0.00 | 0.045 |
| (*Gammaproteobacteria*) | *Nitrosomonadaceae* | **1.33** | 0.91 | 0.019 |
| | *Solimonadaceae* | **0.04** | 0.01 | 0.050 |
| (*Deltaproteobacteria*) | *Sandaracinaceae* | 0.16 | **0.49** | 0.013 |
| | *Ilumatobacteraceae* | 0.86 | **1.72** | 0.041 |
| | *Micrococcaceae* | **1.21** | 0.50 | 0.007 |
| | *Nocardioidaceae* | 1.36 | **2.03** | 0.028 |
| *Actinobacteria* | *Microbacteriaceae* | 0.42 | **0.93** | 0.028 |
| | *Glycomycetaceae* | 0.04 | 0.00 | 0.005 |
| | *Mycobacteriaceae* | 0.24 | **0.60** | 0.010 |
| | *Solirubrobacteraceae* | 1.18 | **1.97** | 0.049 |
| *Acidobacteria* | *Solibacteraceae* | **1.79** | 1.19 | 0.019 |
| | *Pyrinomonadaceae* | **1.05** | 0.53 | 0.004 |
| *Bacteroidetes* | *Microscillaceae* | 0.88 | **2.12** | 0.019 |
| *Gemmatimonadetes* | *Gemmatimonadaceae* | **7.59** | 3.87 | 0.007 |
| *Chloroflexi* | *Roseiflexaceae* | **0.47** | 0.19 | 0.001 |
| | *Anaerolineaceae* | **0.28** | 0.09 | 0.001 |
| *Nitrospirae* | *Nitrospiraceae* | **0.56** | 0.31 | 0.005 |

**Table 5.** Average percentage of families that differs in Voronic (V.) Vorony-Calcic (V.c.) and Grey-Luvic (G.l.) Chernozems. Bigger values and significance levels (*p*) for statistically significant differences given in bold.

| Phyla (Class) | Familia | V.—V.c. | | | V.—G.l. | | | V.c.—G.l. | | |
|---|---|---|---|---|---|---|---|---|---|---|
| | | %V. | %V.c. | *p* | %V. | %G.l. | *p* | %V.c. | %G.l. | *p* |
| (Alpha-Proteobacteria) | *Sphingomonadaceae* | 4.11 | **7.78** | **0.012** | 4.11 | **7.89** | **0.012** | 7.78 | 7.89 | 0.937 |
| | *Azospirillaceae* | 0.12 | **0.52** | **0.035** | 0.12 | 0.20 | 0.065 | 0.52 | 0.20 | 0.065 |
| | *Acetobacteraceae* | 0.08 | **0.22** | **0.001** | 0.08 | **0.29** | **0.001** | 0.22 | 0.29 | 0.180 |
| (Gamma-Proteobacteria) | *Xanthomonadaceae* | 0.75 | **1.93** | **0.024** | 0.75 | **1.45** | **0.030** | 1.93 | 1.45 | 0.485 |
| | *Rhodanobacteraceae* | 0.21 | **0.52** | **0.004** | 0.21 | **0.46** | **0.004** | 0.52 | 0.46 | 0.589 |
| | *Pseudomonadaceae* | 0.30 | 0.43 | 0.343 | 0.30 | **0.57** | **0.024** | 0.43 | 0.57 | 0.589 |
| Actinobacteria | *Geodermatophilaceae* | 0.41 | **1.04** | **0.024** | 0.41 | **0.93** | **0.030** | 1.04 | 0.93 | 0.818 |
| | *Propionibacteriaceae* | 0.23 | 0.14 | 0.282 | **0.23** | 0.05 | **0.008** | **0.14** | 0.05 | **0.039** |
| | *Kineosporiaceae* | 0.09 | 0.28 | 0.240 | 0.09 | **0.39** | **0.004** | 0.28 | 0.39 | 0.240 |

## 4. Discussion

A way to get deeper into the differences between bacterial communities of arable and nonarable soils is to analyze the properties of groups that change their shares in them during long-term agricultural use. Properties that are always known at least to some extent for cultivated microorganisms are relation to oxygen, available range of temperature and pH, as well as the substrates known to be consumed by them. The latter property should be the main one, but often the available information does not cover all the substrates of interest.

*Gemmatimonadaceae*, the most numerous family that prefer conditions of arable Chernozem, possess only few classified genera and species [37]. Genus *Gemmatimonas* that dominates among this family in the samples was initially described by the species *G. aurantiaca*, isolated by cultivation on a low-nutrient medium [38]. This aerobic bacterium can utilize yeast extract, polypepton, succinate, acetate and other organic substrates. Ability to grow on acetate in the minimal medium is of interest as usage of important end product of biological oxidation of organic nutrients. The strain described had grown in a pH range of 6.5–9.5 that includes the values proper to nonacidified Chernozem. Similar pH ranges and lower temperature limits of 15–20 °C are characteristic of other members of the genus as well. Facultatively phototrophic microaerophile *G. phototrophica* grows on yeast extract but weakly on peptone and does not grow on starch [39]. Facultatively phototrophic aerobe *G. groenlandica* consumes yeast extract and glucose, but not arabinose and xylose [40]. It was shown that *Gemmatimonadaceae* in soils prefer less moisture and more alkaline pH and can become the largest group in some soil microbiomes [41].

Among *Solibacteraceae*, another family more numerous in arable soils, genus *Bryobacter* (0.38–1.75%), prevails in the samples. It comprises the only classified species, *B. aggregatus*. It is an aerobe that grows at 4–33 °C and pH 4.5–7.2 [42] on some mono- and heteropolysaccharides, galacturonic and glucuronic acids. *B. aggregatus* was firstly discovered in peat, where it was feeding on the products of *Sphagnum* mosses' decomposition.

It is expected that in arable Chernozem, there is an increased content of *Nitrospiraceae* that oxidizes ammonium [43]. This probably reflects their role in the processing of ammonium fertilizers, primarily organic compounds, with the formation of nitrite that is rapidly oxidized to nitrate by other groups of bacteria. The need to oxidize ammonium nitrogen to nitrate absorption by plants determines the importance of *Nitrospiraceae* for agriculture. Regarding the samples, most of them belong to the well-known [44] genus *Nitrospira*.

In turn, nitrite oxidizers [45] in the analyzed chernozems are represented mainly by the *Nitrosomonadaceae* family, which is also more numerous in conditions of agricultural treatment. Most of *Nitrosomonadaceae* in the samples belong to unclassified groups named Ellin6067 and MND1.

So, this short list of bacterial groups that attract attention by their multiplicity or obvious usefulness shows the benefit from systematic agricultural use of the soil firstly for some organic matter oxidizers and participants in the nitrogen cycle converting ammonium nitrogen. Probably, it is worth putting apart a strictly anaerobic family *Anaerolineaceae*, whose advantage on arable land obviously correlates with the rupture of the soil structure and decrease of aeration due to tamping by agricultural machines.

A similar list of groups preferring untilled Chernozems looks somewhat more diverse despite the size of each of them not being so large. *Microscillaceae* (by Silva database) were represented in the samples by genera *Ohtaekwangia* and *Chryseolinea*, that also are classified in the family *Fulvivirgaceae* by the NCBI [37]. Both genera are obligatorily aerobic. They comprise two classified species each. *O. koreensis* and *O. cribbensis* were isolated by cultivation in aerobic conditions on agar plates covered by cells of *Escherichia coli* but they can also grow in strictly oligotrophic conditions. They can use dextrin and glucose; *O. cribbensis* has broader range of mono- and disaccharides to consume. Acceptable pH range starts at 5.5 for both and extends to alkaline values; temperature toleration starts from 10 °C [46]. *Ch. serpens* grow at pH 5.6–7.7, starting from 13 °C. They can use plenty of mono- and disaccharides, xylan, pectin and starch, but not CM-cellulose [47]. *Ch. soli* tolerates pH 5.5–8.0 and as low as 10 °C, feeding on glucose, arabinose, maltose, saccharose and some other carbohydrates [48].

*Nocardioidaceae* in the samples mainly belong to genus *Nocardioides* that is rich with species. They grow aerobically at mesophilic or psychrophilic conditions, utilizing a wide range of carbon and nitrogen sources, but only a few can hydrolyze cellulose or chitin [49].

The family *Ilumatobacteraceae* shows predominance of genus *Ilumatobacter* in nonarable soils and of an unclassified group named CL500-29 in arable ones. Of the three classified species, two are able to grow in the conditions of soils of middle latitudes, tolerating pH in a range of 6–8, 6–10 and temperatures from 10–12 °C [50]; the third species is too cold-sensitive [51]. They are aerobic and were isolated on marine agar containing mineral salts, peptone and yeast extract.

*Beijerinckiaceae* in the samples mainly comprise representatives of the genus *Microvirga*. They includes several dozens of species rather different by their physiology, many of them were isolated from soils. They are aerobic and typically grow at 16 °C and higher [52], though psychrotolerant a species is known [53], and a thermotolerant one as well [54]. Characteristic is the ability to carry out denitrification and thus to a certain extent compete for nitrate with plants. It would seem that one could expect an increase in the proportion of *Microvirga* simultaneously with the proportion of nitrifiers, such as *Nitrospiraceae* and *Nitrosomonadaceae*, but the opposite effect is observed.

One of the rather small groups—but despite this particularly interesting—is the family *Sandaracinaceae*. The single classified species, for now, is *Sandaracinus amylolyticus* [55]. Acting as a micropredator, it is able to lyse the cells of different organisms including bacteria *E. coli, Klebsiella sp., Staphylococcus aureus, Micrococcus luteus* and yeasts such as *Saccharomyces cerevisiae, Candida albicans* and *Pichia anomala*. On agar plates, it cannot lyse *Nocardia flava, Schizosaccharomyces pombe* and filamentous mold *Mucor hiemalis*. Growth is observed at pH 5.5–8.5 and at least 18 °C and is obligatorily aerobic. *S. amylolyticus* shows no cellulose or chitin degradation but starch is strongly degraded.

A review of the characteristic features of families and their representatives does not provide accurate and definitive answers, but shows the diversity of the affected groups of bacteria, and allows us to raise questions about their norm reaction and about metabolic capabilities available to them. The answers could be provided by the development of experimental approaches to the study of soil microbiota and by using comprehensive microbiological research on the main groups of the soil bacteria.

The results obtained by us are quite consistent with the data on the taxonomic structure of the communities of Voronic Chernozem obtained in other studies [17,18]. At the same time, they are completely different from them regarding the response of the community to the agricultural use of the soil. However, their results are completely different from each

other, so we have to rely on our data and continue the research, keeping the sampling area and method of analysis unchanged to ensure reproducibility of the data.

## 5. Conclusions

In the present study, we revealed that despite agricultural usage appearing to be a somewhat weaker factor in shaping the bacterial community than the chernozem subtype, it concerns more taxonomical groups, the main of which are characteristic organic matter oxidation and participation in the nitrogen cycle. Some of them are capable of biodegradation of starch or pectin but usually not cellulose, chitin or lignin; this may be due to the advantage of fungi in the development of such niches. Among affected bacterial groups, and generally in the communities, we can mainly see consumers of monosaccharides and other small molecules formed under the action of exoenzymes of other organisms. Some consume the end metabolites of syntrophically connected groups, namely acetate and succinate. Herewith, in many cases, the ability of newly described bacteria to use such substances is still out of consideration.

The structure of bacterial communities of the three subtypes of Chernozem considered in this paper predictably turned out to be principally similar, and plowing does not produce a revolution in their common outline. At the same time, the observed differences, considered against the background of a single stable system of intergroup interactions, have a chance to procure more accurate explanations as understanding of it deepens. This task remains all the more urgent because the ecological and economic roles of the most massive groups of soil bacteria may appear only partially understood.

In this study, we obtained a list of families and some conclusions about genera and species of bacteria that respond to long-term arable farming on the Chernozems of the eastern European forest-steppe. For some of them, we were able to trace the characteristic features according to the literature data, apparently related to their main ecological roles in soil microbial communities. The result provides us the opportunity to deepen our research, focusing on studying the factors that determine the population of groups that are potentially important for maintaining the fertility of arable soils.

**Supplementary Materials:** The following supporting information can be downloaded at: https://www.mdpi.com/article/10.3390/d15020191/s1, Figure S1: Shares of bacterial families in microbiota of chernozem. Figure S2: Shares of bacterial genera in microbiota of chernozem.

**Author Contributions:** Conceptualization, K.S.B., V.V.A. and I.V.B.; methodology, K.S.B. and V.V.A.; selection of sampling points, V.A.N. and O.S.B.; experimental procedures, K.S.B., Z.W., O.Y.O., M.V.K. and V.V.K.; formal analysis, V.V.A.; writing—original draft preparation, K.S.B.; writing—review and editing, V.V.A. and I.V.B.; visualization, Y.N.K.; supervision, K.S.B.; project administration, I.V.B.; funding acquisition, K.A.D. and O.A.M. All authors have read and agreed to the published version of the manuscript.

**Funding:** This research was funded by The Russian Science Foundation (RSF), grant number 22-14-20024, and by The Government of the Belgorod oblast, Russia, in the frame of the same grant.

**Institutional Review Board Statement:** Not applicable.

**Informed Consent Statement:** Not applicable.

**Data Availability Statement:** Data is contained within the article.

**Acknowledgments:** The authors acknowledge the support from LLC Rusagro Invest and Belogorye State Nature Reserve, Belgorod oblast, Russia.

**Conflicts of Interest:** The authors declare no conflict of interest.

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
