# Peer review of "The Effect of Long-Term Agricultural Use on the Bacterial Microbiota of Chernozems of the Forest-Steppe Zone"

_diversity, doi:10.3390/d15020191_

Round 1

Reviewer 1 Report

The peer-reviewed manuscript is interesting, providing a basis for further research.

 Lines 65-67: It would be worth formulating a research hypothesis or emphasizing the expectations and results of the conducted research.

 Lines 69-75: Section 2.1 Sample collection - maybe it would be worth putting a map with marked research areas. It would certainly be useful to provide more information on research plots, although some (albeit in my view not exhaustive) information is included in section 3.1. Selection of sample sites. I believe that the information provided in section 3.1. Sampling sites should be moved to section 2.1. retitled, such as this: Sampling sites and sample collection.  It is also worth providing information on the types of chernozems in research sites. Information on the research area could be placed in a table, which would facilitate the reading of an otherwise interesting text.

 Line 127: The description of Figure 2 is very sparing. It would be worth considering providing more information, which would make it easier to read

Author Response

«Lines 65-67: It would be worth formulating a research hypothesis or emphasizing the expectations and results of the conducted research.»

This part of the text has been changed and expanded with a description of the background of the study, its goals, prospects and expectations.

«Lines 69-75: Section 2.1 Sample collection - maybe it would be worth putting a map with marked research areas.»

The map was added as Figure 1. It shows the sampling area in the context of its geographical location and geographical distribution of soil types.

«It would certainly be useful to provide more information on research plots, although some (albeit in my view not exhaustive) information is included in section 3.1. Selection of sample sites. I believe that the information provided in section 3.1. Sampling sites should be moved to section 2.1. retitled, such as this: Sampling sites and sample collection. It is also worth providing information on the types of chernozems in research sites. Information on the research area could be placed in a table, which would facilitate the reading of an otherwise interesting text.»

Former section 3.1 now is included into completely revised and renamed section 2.1. It is also supplemented with data on the climatic features of the area and on experimental design in the text and on the number of certain samples in the form of Table 1.

«Line 127: The description of Figure 2 is very sparing. It would be worth considering providing more information, which would make it easier to read»

Caption of former Figure 2 which now is Figure 3 is detailed as well as captions of other figures.

Thank you for your careful reading and helpful comments.

Reviewer 2 Report

Major Comments:

1.     The whole introduction section lacks direction. What does the study address? Agricultural management systems or tillage or fertilizer input or chernozem-microbiome interactions?  Hypothesis and objectives on the study are not clear. Please indicate the basis for your hypothesis.

2.     Give more information about the experimental design?  How were the chernozems chosen? How many of them? How many replications from each plot?

3.     The plots and figures are poorly described and labelled. Please improve them.  For example, figure 3. Are those error bars? Please change the scale to properly represent all taxonomic groups.

Minor Comments:

Line 60-61: This is a broad statement and is not completely true. There are numerous studies addressing bacterial community changes due to agricultural use.

Line 61: Why mention “micromycetes”. Your study does not address this group.

Line 80:  What primer sets were used for amplification?

Line 81: please change “encode” to “barcode”

Line 104: Move section 3.1 to the methods.

Section 3.2: More descriptions needed for figures. What does figure 1 and 2 imply? Are they scatter plots for richness and evenness?

Line 128: What metrices were used for the beta diversity estimates? This should be clearly indicated in the manuscript. Did you perform ordination plots for beta diversity? Would be good to include that in the manuscript.

Author Response

«1.     The whole introduction section lacks direction. What does the study address? Agricultural management systems or tillage or fertilizer input or chernozem-microbiome interactions?  Hypothesis and objectives on the study are not clear. Please indicate the basis for your hypothesis. »

This part of the text has been changed and expanded with a description of the background of the study, its goals, prospects and expectations.

«2.     Give more information about the experimental design?  How were the chernozems chosen? How many of them? How many replications from each plot? »

Former section 3.1 now is included into completely revised and renamed section 2.1. It is also supplemented with data on the climatic features of the area and on experimental design in the text and on the number of certain samples in the form of Table 1. The map was added as Figure 1. It shows the sampling area in the context of its geographical location and geographical distribution of soil types.

«3.     The plots and figures are poorly described and labelled. Please improve them.  For example, figure 3. Are those error bars? Please change the scale to properly represent all taxonomic groups. »

Former Figure 3 named Figure 4 is revised, its caption is detailed as well as other captions.

«Minor Comments:

Line 60-61: This is a broad statement and is not completely true. There are numerous studies addressing bacterial community changes due to agricultural use. »

The introduction has been expanded and supplemented with a review of works that included similar studies.

«Line 61: Why mention “micromycetes”. Your study does not address this group. »

In the updated version of the introduction, this information is omitted.

«Line 80:  What primer sets were used for amplification? »

Primer sequences are added into section 2.2.

«Line 81: please change “encode” to “barcode” »

This correction has been made.

«Line 104: Move section 3.1 to the methods. »

Former section 3.1 now is included into completely revised and renamed section 2.1.

«Section 3.2: More descriptions needed for figures. What does figure 1 and 2 imply? Are they scatter plots for richness and evenness? »

Caption of former figures 2 and 3 which now are figures 3 and 4 are detailed as well as captions of other figures.

«Line 128: What metrices were used for the beta diversity estimates? This should be clearly indicated in the manuscript. Did you perform ordination plots for beta diversity? Would be good to include that in the manuscript. »

This information has been added to the section 2.3.

Thank you for your careful reading and helpful comments.

Reviewer 3 Report

Dear Authors, I have reviewed the paper " The effect of long-term agricultural use on the bacterial micro-biota of chernozems of the forest-steppe zone  ". The aims of the paper are germane with Diversity topic. The paper is written with a moderate English level. The contribution of this paper to the scientific knowledge is moderate. In my opinion there some flaws and I suggest the corrections in the comments for the authors and also in the file attached.

Author Response

«Dear Authors, I have reviewed the paper " The effect of long-term agricultural use on the bacterial micro-biota of chernozems of the forest-steppe zone  ". The aims of the paper are germane with Diversity topic. The paper is written with a moderate English level. The contribution of this paper to the scientific knowledge is moderate. In my opinion there some flaws and I suggest the corrections in the comments for the authors and also in the file attached.

the aim  of the work carried out is not clearly stated, please specify»

This part of the text has been changed and expanded with a description of the background of the study, its goals, prospects and expectations.

«How large is the area?Please specify»

Length of sampling areas now is specified in 2.1.

«3.1 soil subtype    From what data base did you extrapolate these data?

please specify where are the areas located? »

Former section 3.1 now is included into completely revised and renamed section 2.1. It is also supplemented with data on the climatic features of the area and on experimental design in the text and on the number of certain samples in the form of Table 1. The map was added as Figure 1. It shows the sampling area in the context of its geographical location and geographical distribution of soil types.

«please, implemented the conclusion. It is necessary  a link of the conclusions with the aim of the work. it would also be appropriate to specify what practical application on land management could this study have »

The conclusion is supplemented with a paragraph about the achieved result and prospects of the study.

Thank you for your careful reading and helpful comments.